# Adjustable-Loop Cortical Suspensory Fixation Results in Greater Tibial Tunnel Widening Compared to Interference Screw Fixation in Primary Anterior Cruciate Ligament Reconstruction

**DOI:** 10.3390/medicina58091193

**Published:** 2022-09-01

**Authors:** Tae-Jin Lee, Ki-Mo Jang, Tae-Jin Kim, Sang-Min Lee, Ji-Hoon Bae

**Affiliations:** 1Department of Orthopaedic Surgery, Korea University Guro Hospital, Korea University College of Medicine, Seoul 08308, Korea; 2Department of Orthopaedic Surgery, Korea University Anam Hospital, Korea University College of Medicine, Seoul 02841, Korea

**Keywords:** anterior cruciate ligament, reconstruction, bone tunnel widening, adjustable-loop device, interference screw, hamstring tendon, autograft

## Abstract

*Background:* Although the use of adjustable-loop suspensory fixation has increased in recent years, the influence of the shortcomings of suspensory fixation, such as the bungee-cord or windshield-wiper effects, on tunnel widening remains to be clarified. *Hypothesis/Purpose:* The purpose of this study was to compare adjustable-loop femoral cortical suspensory fixation and interference screw fixation in terms of tunnel widening and clinical outcomes after anterior cruciate ligament reconstruction (ACLR). We hypothesized that tunnel widening in the adjustable-loop femoral cortical suspensory fixation (AL) group would be comparable to that in the interference screw fixation (IF) group. *Methods:* This study evaluated patients who underwent primary ACLR at our institution between March 2015 and June 2019. The femoral and tibial tunnel diameters were measured using plain radiographs in the immediate postoperative period and 2 years after ACLR. Tunnel widening and clinical outcomes (Lysholm score, 2000 International Knee Documentation Committee subjective score, and Tegner activity level) were compared between the two groups. *Results:* There were 48 patients (mean age, 29.8 ± 12.0 years) in the AL group and 44 patients (mean age, 26.0 ± 9.5 years) in the IF group. Tunnel widening was significantly greater in the AL group than that in the IF group at the tibia anteroposterior (AP) middle (2.03 mm vs. 1.32 mm, *p* = 0.017), tibia AP distal (1.52 mm vs. 0.84 mm, *p* = 0.012), tibia lateral proximal (1.85 mm vs. 1.00 mm, *p* = 0.001), tibia lateral middle (2.36 mm vs. 1.03 mm, *p* < 0.001), and tibia lateral distal (2.34 mm vs. 0.85 mm, *p* < 0.001) levels. There were no significant differences between the two groups with respect to femoral tunnel widening and clinical outcomes. *Conclusions:* Tibial tunnel widening was significantly greater in the AL group than in the IF group at 2 years after primary ACLR. However, the clinical outcomes in the two groups were comparable at 2 years.

## 1. Introduction

Anatomical anterior cruciate ligament (ACL) reconstruction (ACLR) is being performed more frequently than in the past, and the use of adjustable- or fixed-loop devices has increased with the development of suspensory fixation. With suspensory fixation, the graft has full contact within the bone tunnel without any foreign material, which may allow early graft integration. However, femoral cortical suspensory fixation may increase the risk of tunnel widening due to micro-movements at the bone-tendon interface, such as the bungee-cord and windshield-wiper effects along the longitudinal and transverse axes, respectively [1,2].

Micro-movements at the tendon-bone interface, such as the bungee-cord and windshield-wiper effects, have resulted in tunnel widening in experimental animal studies [3]. A recent meta-analysis revealed greater femoral tunnel widening with suspensory fixation using a fixed-loop device for ACLR than that with transfemoral cross-pin fixation [4]. In comparison, interference screw fixation has the advantage of reducing graft movement in the tunnel and synovial fluid influx, although there is a risk of graft and tunnel damage during screw insertion. A recent study also reported that tunnel widening with interference screws was less than that with cortical suspensory fixation [5,6].

The use of adjustable-loop devices has increased to compensate for the micro-movement due to cortical suspensory fixation. Several studies have reported that adjustable-loop devices can reduce micro-movements compared to fixed-loop devices after ACLR [7,8,9,10]. However, studies have reported that the adjustable-loop device may cause tunnel widening after surgery [11,12].

Although most previous studies did not report an association between tunnel widening and adverse clinical outcomes, large tunnels may compromise graft fixation during revision surgery or may necessitate two-stage surgery [13,14,15,16,17]. Therefore, the purpose of our study was to compare tunnel widening and clinical outcomes after ACLR with adjustable-loop femoral cortical suspensory fixation to those with interference screw fixation. We hypothesized that tunnel widening in the adjustable-loop femoral cortical suspensory fixation (AL) group would be comparable to that in the interference screw fixation (IF) group.

## 2. Methods

### 2.1. Participants

This study retrospectively analyzed data obtained from our prospective longitudinal observational study of primary ACLR between March 2015 and June 2019 at our institutions, namely Korea University Anam Hospital and Korea University Guro Hospital, Seoul, Republic of Korea. All patients were provided written information about the study and informed consent was obtained. Ethical approval was obtained from the institutional review board (IRB) of our hospital (IRB No. 2021GR0105). Patients who underwent primary ACLR were included, and demographic data were collected from electronic medical records.

Patients who met the following inclusion criteria were included: (1) anatomical single-bundle ACLR using hamstring-tendon autograft (outside-in technique); (2) the same tibial fixation technique (interference screw fixation with additional fixation); and (3) a follow-up of at least 2 years. The exclusion criteria were as follows: (1) other ligament injuries requiring surgical treatment, (2) previous knee surgeries, and (3) subsequent injuries (ACL graft failure, meniscus or cartilage injury, or contralateral ACL rupture).

Of the 352 patients who underwent primary ACLR in our institution, 109 met the inclusion and exclusion criteria for the study. The study was conducted on 92 patients who completed serial tests. The AL group included 48 patients and 44 patients were allocated to the IF group. The patient’s flow diagram is shown in Figure 1.

### 2.2. Surgical Technique and Rehabilitation

All patients underwent anatomical single-bundle ACLR. Surgeries were performed by two senior surgeons. One surgeon used an adjustable-loop cortical suspensory device for femoral fixation, and the other surgeon used an interference screw for femoral fixation. Tibial fixation was performed in the same way in both groups. If there was a meniscus tear, concomitant meniscus repair or a meniscectomy was performed, according to meniscal repairability, prior to ACLR. 

A two-incision outside-in technique was used for anatomical femoral tunnel placement in all cases. In the AL group, the femoral part of the graft was prepared using adjustable-loop suspensory devices (TightRope [Arthrex, Naples, FL, USA]), and the femoral tunnel was made by retrograde drilling. In the interference screw fixation group (IF group), antegrade drilling and bioabsorbable interference screw (GENESYS Matryx [ConMed, Utica, NY, USA]) were used for femoral fixation. In both groups, a bioabsorbable interference screw was used for tibial fixation after cyclic loading. Additionally, a 6.5-mm cancellous screw and spiked washer (Arthrex, Naples, FL, USA) were used on the tibial side. The femoral and tibial tunnels were of the same size as the graft diameter, and bioabsorbable interference screws of the same size were used. In both group, grafts were inserted up to the distal end of the tibial tunnel. For the femoral tunnel, the suspensory group inserted the grafts into the femoral tunnel 5 mm away from proximal end and the interference group inserted the grafts up to the proximal end of the femoral tunnel (Figure 2).

After ACLR, all patients underwent the same rehabilitation protocol. A return to previous sports activities was allowed if the patient achieved >80–90% muscle strength compared with that of the contralateral uninjured leg. 

### 2.3. Radiographic Evaluation

Anteroposterior (AP) and lateral radiographs were taken immediately after surgery and at 12 and 24 months after surgery. On the AP and lateral radiographs, the diameters of the femoral and tibial tunnels were measured between the two sclerotic margins perpendicular to the longitudinal axis at the proximal, middle, and distal sites of the tunnel (Figure 3) [11]. Tunnel widening was calculated by subtracting the diameter measured immediately after surgery from that measured 24 months after surgery.

All radiographic measurements were performed on the picture archiving and communications system (PACS [INFINITT Healthcare, Seoul, Korea]) using a mouse cursor with automated distance calculation. Two orthopedic surgeons, who were trained in identifying each variable, independently reviewed the radiographs. They performed measurements on two separate occasions, 4 weeks apart. 

### 2.4. Clinical Evaluation

All patients were evaluated during their regular visits to the outpatient clinic at 6, 12, and 24 months after surgery. The independent staff (experienced athletic trainers and physical therapists) from our sports rehabilitation center conducted all the tests and obtained the data. Patients completed three patient-reported outcomes, including the Lysholm Knee Scoring Scale, the 2000 International Knee Documentation Committee (IKDC) subjective scale, and Tegner activity level [18,19].

### 2.5. Second-Look Arthroscopy and Graft Evaluation Method

Second-look arthroscopy was performed for skin irritation from the tibial fixation screw at least one year after ACLR. During the second-look arthroscopy, newly identified problems that were not present at the time of ACLR were investigated. In addition, a modified graft maturation scoring system (Korea University Medical Center score; KUMC score) was used for graft evaluation based on previous studies [20,21].

Two blinded observers (orthopedic surgeons) scored each parameter, and interobserver and intraobserver reliabilities were evaluated. Total graft maturation scores (KUMC score) and individual parameter scores were compared between the two groups.

### 2.6. Statistical Analysis

In a pilot study of 10 patients from both groups, tunnel widening was measured, and the mean and standard deviation were calculated. To achieve a power of 80% using a two-group *t*-test with a two-sided significance level of *p* < 0.05, a sample size of 44 in each treatment group was required when calculated using the value measured at the middle of the lateral tibial view as a variable. Data for the final follow-up were available for 48 patients in the AL group and 44 patients in the IF group. Therefore, a power of 80% was reached.

Demographic variables (including patient sex, age, and body mass index), clinical outcomes, and tunnel widening calculated using radiologic measurements were compared between the two groups. Independent *t*-tests and Mann–Whitney U tests were used for continuous variables. Chi-square tests and Fisher’s exact tests were used for categorical variables. Paired-samples *t*-tests and Wilcoxon signed-rank tests were applied to continuous variables within each group.

For each variable, the intraclass and interclass correlation coefficients were calculated to quantify the agreement between the measurements. Correlation coefficients >0.75 and <0.4 represent good and poor agreement, respectively [22]. In addition, all variables were expressed using mean values and standard deviations.

Statistical significance was set at *p* < 0.05. Statistical analysis was performed using SPSS for Windows (version 20.0; IBM Corp., Armonk, NY, USA).

## 3. Results

Demographic data of the 92 patients are presented in Table 1. No significant differences in age, sex, or body mass index were observed between the two groups.

### 3.1. Radiologic Outcomes

Significant tunnel widening was observed at all measured sites between the immediate postoperative period and 2 years after ACLR in both groups (Table 2 and Table 3). All correlation coefficients ranged from 0.813 to 0.935, indicating good reliability.

The tunnel widening in the AL group was significantly greater than that in the IF group at the tibia AP middle, tibia AP distal, tibia lateral proximal, tibia lateral middle, and tibia lateral distal (Table 4).

### 3.2. Clinical Outcomes

There were no significant differences in patient-reported outcomes (Lysholm score, 2000 IKDC subjective score, and Tegner activity level) between the two groups at 2 years after ACLR (Table 5).

### 3.3. Second-Look Arthroscopic Evaluation

Twenty-four patients in each group underwent second-look arthroscopic evaluation, and the operation was performed at an average of 24.5 and 26.8 months after ACLR in the AL and IF groups, respectively. After the second-look arthroscopic examination, total graft maturation scores (KUMC score) were compared between the two groups. In patients undergoing second-look arthroscopy, there was no significant difference in the KUMC scores between the two groups (Table 6). All correlation coefficients ranged from 0.838 to 0.905, indicating good reliability. Regarding newly identified problems in group 1, two meniscal tears were identified. In group 2, two cyclops lesions and two meniscal tears were identified.

## 4. Discussion

The main findings of this study were that ACLR using interference screw fixation was associated with less postoperative tibial tunnel widening than that using adjustable-loop femoral cortical suspensory fixation at the 2-year follow-up and that there were no significant differences between the two groups with regard to femoral tunnel widening or clinical outcomes. However, it was difficult to clearly explain the difference in tunnel widening between the two groups. The exact reason was not found, but various factors are presumed to be involved.

Several potential reasons for the differences between the two groups can be considered with regard to tibial tunnel widening. Interference screw fixation has the advantage of reducing micro-movements compared to adjustable-loop femoral cortical suspensory fixation. In a randomized controlled trial, Fauno and Kaalund [14] found that tunnel widening is influenced by the mechanical properties of the implants. They found that extracortical fixation of the graft was associated with more laxity than close-to-joint fixation. Similarly, Giorgio et al. [23] reported that the use of a more elastic fixation system (TightRope) is associated with more laxity and results in greater tunnel widening. They explained that graft motion, such as the bungee-cord and windshield-wiper effects, is caused by low-stiffness systems such as those using extracortical fixation. In the study by Sabat et al. [24], tunnel widening was significantly lesser in the transfix group than in the EndoButton group. They thought that this was because the fixation point was farther away from the aperture. Interestingly, Lind et al. [25] suggested that the femoral fixation technique could potentially affect the biomechanical and biological environment in the tibial tunnel, thus affecting tunnel widening in the tibia. Accordingly, we thought that the difference in the femoral fixation method in our study could have influenced the tunnel widening in the tibia.

However, unlike tibial tunnel widening, a significant difference in femoral tunnel widening was not seen in the present study. We attributed this to the difference between the characteristics of the distal femur and the proximal tibia. Chang et al. [26] reported the mechanical and structural properties of distal femur and proximal tibial bones in vivo. Their study showed that the stiffness of the distal femur was greater than that of the proximal tibia. Therefore, the widening of the femoral tunnel could be expected to be relatively small. Kuskucu [27] reported that a graft that becomes free because of fixation in the tunnel at a site distant from the tunnel opening in the joint will result in greater micro-movement with knee motion. Since micro-movement in the femoral tunnel was less than that in the tibial tunnel, the difference in the femoral tunnel width was not significant.

In contrast to the results of our study, some previous studies have reported that greater femoral tunnel widening occurred when using a suspensory device on the femoral side. Mayr et al. [5] reported that ACLR using adjustable-loop cortical button fixation was associated with greater femoral tunnel enlargement than that with biodegradable interference screw fixation. However, their study differed from our study in that they used an all-inside ACLR. In the suspensory group, patients were treated with all-side ACLR using adjustable-loop button fixation on both the femoral and tibial sides. Suspensory tibial fixation may have influenced whether tunnel widening occurred in the femur or tibia. In a similar study, Baumfeld et al. [15] reported a comparison between double cross-pin and suspensory graft fixation. Femoral tunnel widening associated with the use of the EndoButton suspensory fixation system was significantly greater than that with the use of double cross-pins for fixation within the tunnel. However, their results may be different from ours because they did not standardize the tibial fixation method.

In this study, both groups showed excellent overall clinical outcomes. Patient-reported outcomes, including the Lysholm score, 2000 IKDC subjective score, and Tegner activity level, improved over a postoperative period of 2 years. However, there were no significant differences between the two groups. In addition, in patients undergoing second-look arthroscopy, there was no significant difference in the graft maturation scoring system (KUMC) scores between the two groups. Raj et al. [28] evaluated the correlation between the bone tunnel diameter following ACLR using a hamstring-tendon autograft and functional outcomes. They concluded that neither the diameter nor widening of the bone tunnel during the follow-up period was correlated with functional outcomes. Several studies have reported that there were no clinical implications of tunnel widening after ACLR [12,29,30,31,32]. However, the absence of clinical differences may be attributed to the small difference in tunnel widening between the two groups. Therefore, the results of long-term follow-up may vary. Tunnel widening should be considered in revision ACLR, but a difference of approximately 1 mm will not have significant implications.

This study has several limitations. First, this study was designed retrospectively, and the operations were performed by two surgeons. Although the same outside-in technique was used, the difference between antegrade and retrograde drilling for the femoral tunnel could affect the results. Second, the use of an interference screw can affect the width of the tunnel. There was a risk of graft and tunnel damage during screw insertion. However, the analysis showed that there was no significant difference in tunnel widening by the interference screw in the present study. Third, the minimum follow-up period was only 2 years. Although there was no difference in clinical outcomes in this study, further studies are needed as outcomes may vary at the mid- to long-term follow-up.

## 5. Conclusions

In conclusion, tibial tunnel widening after ACLR using a hamstring-tendon autograft was significantly greater with adjustable-loop femoral cortical suspensory fixation than with interference screw fixation at the 2-year follow-up. However, the clinical outcomes in the two groups were comparable at the 2-year follow-up.

## Figures and Tables

**Figure 1 medicina-58-01193-f001:**
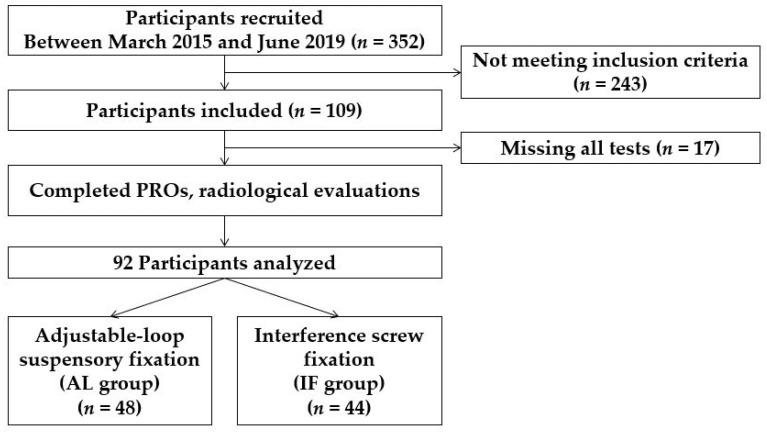
Patient flow diagram. AL, adjustable-loop suspensory fixation; IF, interference screw fixation.

**Figure 2 medicina-58-01193-f002:**
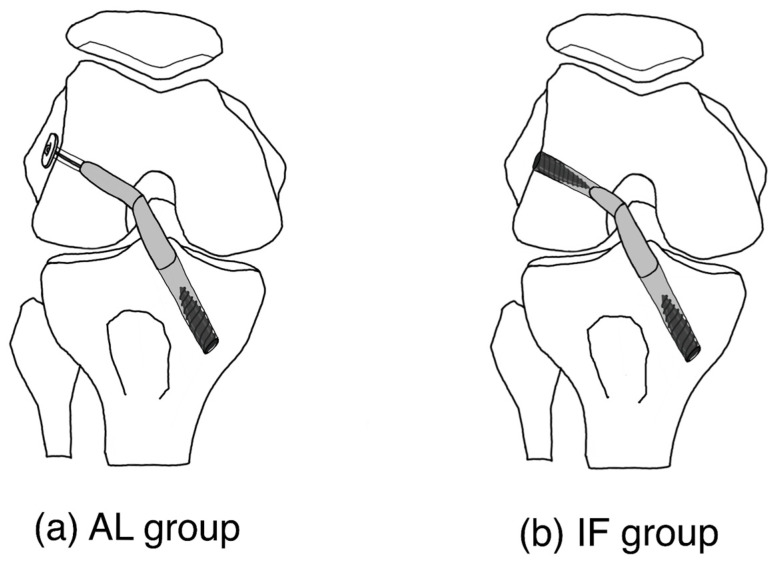
Surgical technique.

**Figure 3 medicina-58-01193-f003:**
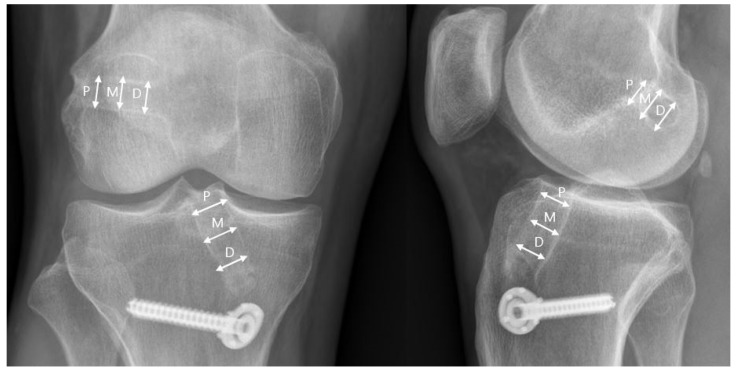
Anteroposterior and lateral radiographs of a right knee showing measurement of each tunnel at proximal (P), middle (M), and distal (D) locations.

**Table 1 medicina-58-01193-t001:** Demographic factors.

	Total (*n* = 92)	AL Group (*n* = 48)	IF Group (*n* = 44)	*p*-Value *
Age (years)	28.0 ± 11.1	29.8 ± 12.0	26.0 ± 9.5	0.081
Sex				0.557
Male, *n* (%)	79 (85.9)	40 (83.3)	39 (88.6)	
Female, *n* (%)	13 (14.1)	8 (16.7)	5 (11.4)	
BMI (kg/m^2^)	25.3 ± 3.3	25.7 ± 3.4	24.9 ± 3.3	0.259

Values are given as mean and standard deviation. BMI, body mass index. * Comparisons between the AL and IF groups. AL, adjustable-loop suspensory fixation; IF, interference screw fixation.

**Table 2 medicina-58-01193-t002:** Immediate and 2-year postoperative tunnel size in the AL group.

	Immediate Postoperative	2-Year Postoperative	*p*-Value *
Femur AP Proximal (mm)	9.10 ± 1.23	9.86 ± 1.70	**<0.001**
Femur AP Middle (mm)	9.16 ± 1.17	10.30 ± 1.82	**<0.001**
Femur AP Distal (mm)	9.17 ± 1.15	10.45 ± 1.85	**<0.001**
Tibia AP Proximal (mm)	9.33 ± 1.05	10.63 ± 1.42	**<0.001**
Tibia AP Middle (mm)	9.53 ± 1.02	11.56 ± 1.60	**<0.001**
Tibia AP Distal (mm)	9.60 ± 1.07	11.12 ± 1.45	**<0.001**
Femur Lat Proximal (mm)	8.64 ± 1.10	9.62 ± 1.88	**<0.001**
Femur Lat Middle (mm)	8.70 ± 1.12	9.98 ± 1.90	**<0.001**
Femur Lat Distal (mm)	8.70 ± 1.09	9.94 ± 1.85	**<0.001**
Tibia Lat Proximal (mm)	9.09 ± 0.88	10.94 ± 1.42	**<0.001**
Tibia Lat Middle (mm)	9.34 ± 1.02	11.70 ± 1.62	**<0.001**
Tibia Lat Distal (mm)	9.33 ± 1.11	11.67 ± 1.61	**<0.001**

Bold indicates statistical significance (*p* < 0.05). Values are given as mean and standard deviation. * Comparisons between the immediate and 2-yearpostoperative values. Lat, lateral. AP, anterior posterior; Lat, lateral.

**Table 3 medicina-58-01193-t003:** Immediate and 2-year postoperative tunnel size in the IF group.

	Immediate Postoperative	2-Year Postoperative	*p*-Value *
Femur AP Proximal (mm)	9.61 ± 1.42	10.54 ± 1.51	**<0.001**
Femur AP Middle (mm)	9.43 ± 1.44	10.79 ± 1.53	**<0.001**
Femur AP Distal (mm)	9.20 ± 1.18	10.76 ± 1.37	**<0.001**
Tibia AP Proximal (mm)	9.48 ± 0.85	10.81 ± 1.39	**<0.001**
Tibia AP Middle (mm)	9.58 ± 0.89	10.89 ± 1.06	**<0.001**
Tibia AP Distal (mm)	9.53 ± 0.84	10.37 ± 1.05	**<0.001**
Femur Lat Proximal (mm)	9.19 ± 1.30	9.74 ± 1.13	**<0.001**
Femur Lat Middle (mm)	8.95 ± 1.09	9.86 ± 1.10	**<0.001**
Femur Lat Distal (mm)	8.79 ± 0.89	9.99 ± 1.02	**<0.001**
Tibia Lat Proximal (mm)	9.38 ± 0.87	10.38 ± 1.17	**<0.001**
Tibia Lat Middle (mm)	9.47 ± 0.82	10.50 ± 0.99	**<0.001**
Tibia Lat Distal (mm)	9.45 ± 0.87	10.30 ± 1.08	**<0.001**

Bold indicates statistical significance (*p* < 0.05). Values are given as mean and standard deviation. * Comparisons between the immediate and 2-year postoperative values. Lat, lateral.

**Table 4 medicina-58-01193-t004:** Tunnel widening between immediate and 2-year postoperative.

	AL Group (*n* = 48)	IF Group (*n* = 44)	*p*-Value *
Femur AP Proximal (mm)	0.76 ± 1.37	0.93 ± 1.59	0.574
Femur AP Middle (mm)	1.14 ± 1.48	1.36 ± 1.46	0.459
Femur AP Distal (mm)	1.27 ± 1.51	1.56 ± 1.45	0.355
Tibia AP Proximal (mm)	1.30 ± 1.33	1.33 ± 1.61	0.914
Tibia AP Middle (mm)	2.03 ± 1.45	1.32 ± 1.34	**0.017**
Tibia AP Distal (mm)	1.52 ± 1.41	0.84 ± 1.09	**0.012**
Femur Lat Proximal (mm)	0.98 ± 1.47	0.54 ± 1.73	0.198
Femur Lat Middle (mm)	0.29 ± 1.59	0.91 ± 1.46	0.244
Femur Lat Distal (mm)	1.23 ± 1.60	1.20 ± 1.16	0.905
Tibia Lat Proximal (mm)	1.85 ± 1.25	1.00 ± 1.19	**0.001**
Tibia Lat Middle (mm)	2.36 ± 1.46	1.03 ± 1.10	**<0.001**
Tibia Lat Distal (mm)	2.34 ± 1.25	0.85 ± 1.27	**<0.001**

Bold indicates statistical significance (*p* < 0.05). Values are presented as mean and standard deviation. * Comparisons between the AL and IF groups. Lat, lateral.

**Table 5 medicina-58-01193-t005:** Clinical outcomes: 2-year postoperative patient-reported outcomes.

	AL Group (*n* = 48)	IF Group (*n* = 44)	*p*-Value *
Lysholm score	82.5 ± 14.5	83.5 ± 19.3	0.766
IKDC subjective score	75.3 ± 17.4	80.5 ± 13.6	0.121
Tegner activity level **	5	6	0.153

Values are presented as mean and standard deviation. * Comparisons between the AL and IF groups. ** Data are presented as a frequency distribution. IKDC, international knee documentation committee.

**Table 6 medicina-58-01193-t006:** Second-look arthroscopic evaluation: graft maturation scores (KUMC scores).

	AL Group (*n* = 24)	IF Group (*n* = 24)	*p*-Value *
Graft integrity **	2	2	1.000
Graft synovial coverage **	2	2	0.690
Graft tension **	2	2	0.931
Graft vascularization **	2	2	0.306
KUMC score **	8	7	0.741

* Comparisons between the AL and IF groups. ** Data are presented as a frequency distribution. KUMC, Korea University Medical Center.

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
