# Peer review of "Adjustable-Loop Cortical Suspensory Fixation Results in Greater Tibial Tunnel Widening Compared to Interference Screw Fixation in Primary Anterior Cruciate Ligament Reconstruction"

_medicina, 2022, doi:10.3390/medicina58091193_

Round 1
Reviewer 1 Report
It is interesting that the authors found there was no difference regarding the widening for femoral tunnel in AL and IF group. However, the difference was found at middle and distal part of tibial tunnel without the significance in clinical results. As surgical technique used in this study utilized interference screw for the tibia, I assume that the micromotion from the femoral fixation was diminished in the tibial tunnel especially at middle and distal part.
Please add the discussion regarding aforementioned result.
Surgical technique
There was a lack of information how long the graft was inserted in each tunnel.
Was femoral tunnel created by retrograde drill or transportal technique? When the cortical suspension device is used, the proximal tunnel should be 4-5mm to avoid the migration of the button. 9mm diameter for the proximal femoral tunnel right after surgery seems very dangerous right after the reconstruction.
Also I want to know if the graft was inserted at middle and distal part of tibia or just open space.
Reviewer 2 Report
The comparison of the techniques used is very interesting and important for the health professional to be able to decide which technique is most suitable for each patient. This manuscript makes a good comparison between the techniques by performing a correct statistical analysis.
The text is well written.
However, I also found some words where spaces are missing, so the text should be revised.
In the subchapter 2.2 where the techniques used are described, I suggest that you put two images to be able to observe the techniques.
When the results are presented in tables 4 and 5, it would be interesting to have more comments since the p-value is quite different.
I have written some writing details that I have been underlining throughout the text.
Very good work, congratulations!

Reviewer 3 Report
thanks for submitting this well-written paper
the topic is relevant even if not entirely new
specific comments:
-please be sure that references are relevant, well-formatted, and up to date
-please be sure that English grammar is perfect, and seek aid from a native speaker if needed
-introduction was well written, please underline what is missing in previous studies and why is this paper needed
-methodology is clear. please report in the statistical section how you repost data (mean? median etc..) and normal distribution test
-line 130-133 please move to the statistical section
-tables please specify in the notes which tests are used
-i suggest to include a figure showing the different tunnel enlargement between groups
Reviewer 4 Report
medicina-1880792_review
Title: Adjustable-loop Cortical Suspensory Fixation Results in Greater Tibial Tunnel Widening Compared to Interference Screw Fixation in Primary Anterior Cruciate Ligament Reconstruction
Comments and Suggestions for Authors
Dear authors,
I was glad to have the opportunity to review this manuscript that aimed to compared tunnel enlargement and clinical outcomes after anterior cruciate ligament reconstruction with adjustable-loop femoral cortical suspensory fixation and interference screw fixation. You hypothesized that tunnel widening in the adjustable-loop femoral cortical suspensory fixation group would be comparable to that in the interference screw fixation group. After the results, you concluded that adjustable loop femoral cortical suspensory fixation group showed a greater tibial tunnel widening than interference screw fixation group at final of the follow-up period, and the clinical outcomes in the two groups were comparable at 2-year follow-up.
In my opinion, the introduction, objectives, methods, results, discussion and conclusions sections are well-described and analyses are appropriate.
I would like to make some minor comments that could be addressed to improve the document, in my opinion.
Specific comments:
Introduction
- In my opinion, the introduction is complete, the objective and hypothesis of the study are adequate.
Material and methods
- Page 2, lines 78-83. Could you add information about the places where this study was carried out, institution, country….?
- Page 2, lines 90-93. How were the participants assigned to each of the groups? What criteria were followed to assign a participant to one group or another? This is essential for your study, please clarify.
- Page 4. Lines 138-140: Please report some information about the Lysholm Knee Scoring Scale, the 2000 International Knee Documentation Committee 139 subjective scale and Tegner activity level if possible. Please add some supporting references, it is necessary.
Results
- The results section is well-structured and comprehensive.
Discussion
-Your discussion section is adequate and complete.
Conclusions
-Your conclusions are appropriate.
I hope that my comments could help to improve the paper. Congratulations for your research.
